# Duration of Immunotherapy in Non-Small Cell Lung Cancer Survivors: A Lifelong Commitment?

**DOI:** 10.3390/cancers15030689

**Published:** 2023-01-22

**Authors:** Carlo Putzu, Stefania Canova, Panagiotis Paliogiannis, Renato Lobrano, Luca Sala, Diego Luigi Cortinovis, Francesca Colonese

**Affiliations:** 1Medical Oncology Unit, University Hospital (AOU) of Sassari, 07100 Sassari, Italy; 2Medical Oncology Unit, Fondazione IRCCS San Gerardo dei Tintori Monza, 20900 Monza, Italy; 3Anatomic Pathology and Histology, University Hospital (AOU) of Sassari, 07100 Sassari, Italy; 4Department of Medicine, Surgery, and Pharmacy, University of Sassari, 07100 Sassari, Italy

**Keywords:** lung, cancer, NSCLC, immunotherapy, chemotherapy, immune checkpoint inhibitors

## Abstract

**Simple Summary:**

Lung cancer is one of the most common human cancers and the leading cause of cancer-related mortality worldwide. The advent of immunotherapy for the treatment of non-small cell lung cancer has significantly improved progression-free survival, overall survival, and the patient’s quality of life in comparison to chemotherapy. Currently, it is estimated that long-term survival can be achieved in more than 15% of patients treated with immunotherapy. Therefore, the optimal duration of immunotherapy in long survivors needs to be established to avoid overtreatment, side effects, and high costs and at the same time, protect them from potential disease relapse. The aim of this review is to discuss all the aspects related to the optimal duration of immunotherapy in long survivors with non-small cell lung cancer.

**Abstract:**

Lung cancer is one of the most common human malignancies and the leading cause of cancer-related death worldwide. Novel therapeutic approaches, like targeted therapies against specific molecular alterations and immunotherapy, have revolutionized in the last decade the oncological outcomes in patients affected by non-small cell lung cancer (NSCLC). The advent of immunotherapy for the treatment of NSCLC has significantly improved overall and progression-free survival, as well as the patient’s quality of life in comparison to traditional chemotherapy. Currently, it is estimated that long-term survival can be achieved in more than 15% of NSCLC patients treated with immunotherapy. Therefore, the optimal duration of immunotherapy in long survivors needs to be established to avoid overtreatment, side effects, and high costs and at the same time, protect them from potential disease relapse or progression. We performed a narrative review to discuss all the aspects related to the optimal duration of immunotherapy in long survivors with NSCLC. Data regarding the duration of immunotherapy in the most impacting clinical trials were collected, along with data regarding the impact of toxicities, side effects, and costs for healthcare providers. In addition, the two-year immunotherapy scheme in patients who benefit from first-line or subsequent treatment lines are examined, and the need for biomarkers that can predict outcomes during and after immunotherapy cessation in patients affected by NSCLC are discussed.

## 1. Introduction

Lung cancer is one of the most common human cancers and the leading cause of cancer-related mortality worldwide, according to data from the Global Cancer Observatory (https://gco.iarc.fr/, accessed on 10 November 2022). Histologically, non-small-cell lung cancer (NSCLC) is the most frequent lung cancer subtype, accounting for approximately 85% of diagnoses, and comprises the most common histotypes like adenocarcinoma and squamous cell carcinoma [1]. About two-thirds of NSCLCs are detected at an advanced or metastatic disease stage [2]. In recent years, the treatment of advanced NSCLC has radically evolved with the introduction in clinical practice of targeted therapies against specific molecular alterations and immune checkpoint inhibitors (ICIs). A clear advantage in survival has been demonstrated for targeted therapies in patients with tumors harboring genetic alterations in driver oncogenes like Epidermal Growth Factor Receptor (EGFR) mutations, Anaplastic Lymphome Kinase (ALK) or c-Ros Oncogene 1 (ROS1) fusions and others [3,4], while two different classes of ICIs to treat non-oncogene addicted tumors have shown similar advantages in clinical trials [5]. The first class includes the inhibitors of programmed cell death protein 1 (PD-1; nivolumab, pembrolizumab) or its ligand (PD-L1; atezolizumab); the second class comprises inhibitors of cytotoxic lymphocyte-associated protein 4 (CTLA-4), like ipilimumab [5].

These medications act by removing specific mechanisms of checkpoint inhibition, which have the role of preventing a harmful immune attack to self-antigens during physiological immune responses; in patients with cancer, removal of such inhibitions enhances immunological responses to cancer [6]. Nevertheless, durable clinical responses are seen only in a subset of patients, while most patients do not respond to treatment or relapse after an initial response. Responses to immunotherapy are shaped by both genetic and environmental factors and by other immunomodulatory treatments such as radiotherapy, chemotherapy, and several targeted therapies [7].

In any case, the advent of immunotherapy has significantly improved progression-free survival (PFS), overall survival (OS), and the patient’s quality of life in comparison to chemotherapy. In pretreated advanced NSCLC, administration of nivolumab or atezolizumab has shown improved clinical outcomes compared to chemotherapy [8]; similarly, Pembrolizumab administered for two years maximum, showed better results in both treatment-naïve and pretreated patients [9,10]. In addition, in the first-line setting, the ipilimumab/nivolumab combination has significantly increased PFS and OS in advanced NSCLC patients followed up for two years [11,12]. Based on these results, the Food and Drug Administration (FDA) and the European Medicines Agency (EMA) approved ICIs for first-line treatment of advanced NSCLC. Currently, several ICIs are recommended by international guidelines in accordance with PD-L1 expression levels (Figure 1, https://www.nccn.org, accessed on 15 January 2023). With the introduction in the daily practice of ICIs and the raising of the 5-year OS rates, the number of long-term survivors has increased, as reported in several clinical trials like CheckMate 003, CheckMate 057 and 017, and KEYNOTE 001 [13,14,15]. In these trials, long-term survival was achieved in more than 15% of patients (vs. 5% in the pre-immunotherapy era); however, the optimal duration of immunotherapy remains to be established. While in earlier studies, the treatment was administered until disease progression or unacceptable toxicity, in more recent trials, an administration limit of two years was introduced. Real-world data on ICIs discontinuation are limited, and only a single randomized phase IIIb/IV trial has been published to date [16].

We performed herein a review of the current literature with the aim of investigating the optimal duration of ICIs treatment in advanced NSCLC, focusing on clinical outcomes, toxicity, healthcare costs, and rechallenge strategies.

## 2. Long-Term Survivorship in Clinical Trials

Numerous trials on immunotherapy in NSCLC have been recently published, with various treatment durations with anti PD-1/PD-L1 drugs (Table 1). In several phase 3 studies, treatment was delivered until the progression of the disease (PD) or, in some cases, until loss of clinical benefit; in others, the treatment administration limit was set at two years or until unacceptable toxicity.

Topalian et al. (CA209-003) reported that pretreated NSCLC patients could obtain a long-term benefit from two years of the administration of nivolumab, with an estimated 5-year survival rate of 16% [13]. A similar long-term survival benefit has been achieved with pembrolizumab administered until disease progression in the KEYNOTE-001 study [17]. In the KEYNOTE 024 study, two years maximum of pembrolizumab treatment was planned; the study showed that pembrolizumab monotherapy significantly increased OS in comparison to standard chemotherapy [18]. The 5-year OS rate in the pembrolizumab group (31.9%) was approximately two-fold the one observed in the chemotherapy group (16.3%), and the median OS was longer more than one year in the pembrolizumab arm (26.3 months) in comparison to the chemotherapy arm (13.4 months; HR, 0.62; 95% CI, 0.48 to 0.81). Twelve patients received pembrolizumab (rechallenge) after PD; among them, eight (66.7%) were alive at the final follow-up. Four patients (33.3%) had an objective response after the second course of pembrolizumab (all of them were partial responders, PRs), while six patients (50%) had stable disease (SD) [19].

In the KEYNOTE 010 study, pembrolizumab continued to provide greater long-term benefit than docetaxel in patients with previously treated advanced NSCLC, with both PD-L1 TPS ≥ 50% and ≥1%. These results have been updated in a subsequent analysis of efficacy and safety outcomes, after approximately five years of follow-up [15]. In this analysis, outcomes were reported for 79 patients who completed 35 cycles/2 years of pembrolizumab and for 21 patients who were rechallenged with pembrolizumab. The 5-year OS rates of pembrolizumab and docetaxel were 25% and 8.2%, respectively, in patients with PD-L1 TPS ≥ 50%, while it was 15.6% and 6.5%, respectively, in those with PD-L1 TPS ≥ 1%. Among the 79 patients who completed 35 cycles/2 years of pembrolizumab, the 3-year OS rate was 83%; among those who were rechallenged with pembrolizumab, eleven (52.4%) had an objective response, and 15 (71.4%) were alive at data cutoff [18,20]. In the CheckMate 078 study, nivolumab was used until disease progression in previously treated advanced NSCLC. This study showed again that immunotherapy with nivolumab improved OS when compared with docetaxel [21].

Recently, a pooled analysis of the phase 3 CheckMate 017 and CheckMate 057 studies was published, including 854 patients who received nivolumab or docetaxel until progression or unacceptable toxicity [14]. The 5-year OS rate was 13.4% with nivolumab and 2.6% with docetaxel; this finding is particularly interesting considering that these studies are those with the longest follow-up time among randomized phase 3 trials testing a PD-1 inhibitor in previously treated advanced NSCLC. After five years, 50 out of 427 patients treated with nivolumab and nine out of 427 patients in the docetaxel arm were still alive. Pooled OS rates at five years were similar regardless of squamous or non-squamous histology, irrespective of tumor PD-L1 expression, and it was observed across several subgroups of patients. Landmark analysis of PFS and OS by progression-free status at two, three, and four years, demonstrated that a consistent number of patients treated with nivolumab remained progression-free during subsequent years. Patients who were progression-free at two, three, and four years had an increased probability (59.6%, 78.3%, and 87.5%, respectively) of still being progression-free at five years, as well as a chance to survive of 82.0%, 93.0%, and 100.0%, respectively. Moreover, in this report, most patients had CRs and PRs as the best treatment responses [14].

Atezolizumab showed similar results in terms of efficacy and long-term responses in comparison to docetaxel in the phase 2 POPLAR and phase 3 OAK studies, performed in patients with previously treated advanced NSCLC [22]. A recent real-world study compared atezolizumab and nivolumab with docetaxel for the treatment of patients with advanced NSCLC who did not respond to platinum-based chemotherapy, providing similar results. The study found that atezolizumab was associated with superior OS compared with docetaxel and similar OS compared with nivolumab. Of note, this was one of the first analyses of OS in a large representative real-world cohort of patients, and its findings provided supportive evidence of the good performance of approved therapies in routine clinical practice [23].

Finally, results are controversial regarding the “chemo-free regimens,” like the association of several ICIs with potential additional effects (PD1 plus CTLA-4 inhibitors). In the Checkmate 227 trial, treatment was allowed until disease progression or unacceptable toxicity or, for the immunotherapy regimens, until two years of follow-up. The study demonstrated a better OS in the experimental arm with nivolumab plus ipilimumab, regardless of tumor PD-L1 expression [12]. On the contrary, in the MISTIC trial, the combination of durvalumab with tremelimumab showed no OS increase compared to platinum-based doublet chemotherapy [24]. In this trial, the duration of the treatment with durvalumab was allowed until disease progression, while tremelimumab was planned for up to four doses. Recently, the combination of PD-1/CTLA-4 blockade plus a short course of platinum-chemotherapy showed encouraging results, as well as a favorable risk–benefit profile versus chemotherapy alone. The combination led to increased tumor response and better median OS (15.6 months in the experimental group versus 10.9 months in the control arm) [25]. Further trials are necessary to draw better conclusions regarding the “chemo-free regimens”.

## 3. Trials Investigating Cessation of Immunotherapy

Currently, most registered phase 3 trials have not been designed either to define an optimal duration of ICI therapy nor to investigate the potential safety hazards of stopping the treatment. The current practice of providing lifetime immunotherapy for advanced or metastatic pretreated NSCLC has been investigated in the CheckMate 153 study, which is the only randomized clinical trial designed for this purpose. The study was a large (1428 patients) phase 3B/4 trial evaluating nivolumab in patients with advanced NSCLC that progressed despite treatment with systemic therapy. The primary endpoint of the study was to evaluate the incidence of treatment-related adverse events (TRAEs), ranging from grade 3 to 5, based on the National Cancer Institute’s Common Terminology Criteria for Adverse Events, version 4.0 [26]. A subset of this cohort (252 patients) was selected for exploratory analysis, focusing on those who had received nivolumab for one year. Patients were randomly assigned to continue nivolumab or to interrupt it. The main objective of the analysis was to explore the impact of treatment duration on oncological efficacy and safety; the specific endpoints were safety, PFS, OS, and ORR assessed one year after randomization [16]. The PFS population included patients who did not have a progressive disease and had not started a new systemic therapy before randomization.

The median PFS was longer for patients who continued the treatment, as opposed to those who ceased it (24.7 months versus 9.4 months, hazard ratio [HR: 0.56; 95% CI: 0.37–0.84]). In addition, median PFS in patients who achieved CR or PR was longer for those who continued the treatment in comparison to those whose treatment was completed after one year (31 months versus 10.6 months; HR: 0.46; 95% CI: 0.27–0.77). On the other hand, no significant difference in median PFS was observed in patients with SD at the time of randomization (11.8 months vs. 9.4 months; HR: 1.01; 95% CI: 0.51–2.01).

Similarly, median OS was longer in both the intention-to-treat (ITT) (not reached vs. 28.8 months; HR: 0.62; 95% CI: 0.42–0.92) and in the PFS population (not reached vs. 32.5 months; HR: 0.61; 95% CI: 0.37–0.99), with continuous versus one-year fixed-duration treatment. At the same time, median OS was longer with continuous therapy compared to one-year, fixed-duration therapy in patients with CR or PR at random assignment (not reached vs. 33.5 months; HR: 0.50; 95% CI: 0.26–0.97), but it was similar in patients with SD (32.2 months vs. 26.6 months; HR: 0.88; 95% CI: 0.42–1.84). Furthermore, median OS was not statistically different in patients with PD at random assignment between continuous therapy and one-year duration therapy (not reached versus 23.8 months with one-year treatment (HR: 0.70; 95% CI: 0.37–1.33).

Regarding safety, patients in the continuous therapy arm had more adverse events when compared with the one-year treatment group (AEs; 32.3 vs. 15.2%), as well as TRAEs (48.0 vs. 26.4%) and TRAEs leading to discontinuation (9.4 vs. 1.6%). However, no new safety concerns were reported. Based on these results, the authors concluded that continuing nivolumab beyond one year provides a meaningful clinical benefit in this setting of patients. Nevertheless, it must be kept in mind that this explorative analysis has several limitations, as reported by the authors themselves. First, the study was not a pre-planned analysis, and therefore, the sample size was not determined in advance. Second, patients were not stratified according to the response at the time of randomization. Third, baseline measurements were not determined at the time of assignment, which means that patient response assessment has a limited value. Finally, randomization was not blinded, and the authors did not plan a blinded independent review, which might have introduced a risk of bias regarding the efficacy of the treatment [27]. Therefore, this exploratory analysis should be considered as an encouraging starting point and hypothesis-stimulating research, while its results should be confirmed in randomized, blinded clinical trials able to provide a greater level of scientific evidence.

Apart from the CheckMate153 trial, other attempts have been made to investigate immunotherapy duration through the collection and analysis of empirical data from daily clinical practice. A few clinical case reports and short retrospective observational case series have been published, including from a minimum of two patients [28] to a maximum of 59 patients [29]. In these reports, all patients (most of them in second- and third-line settings) were treated with a single anti-PD-1 or PDL-1 agent, which in most cases was nivolumab. The main causes of treatment interruption reported were adverse events and rarely socioeconomic reasons [28,29,30,31,32]. Despite the methodological and scientific evidence quality differences between these studies, they globally seem to suggest an improvement of PFS for patients with CR or PR at the time of treatment discontinuation, as well as for those interrupting treatment for AEs. Furthermore, the greatest benefit was generally observed in patients receiving immunotherapy for longer periods (at least 12 months), as underlined by the findings of Kim and colleagues [32]. However, the limitations of these reports, which are retrospective, heterogeneous, and with small patient samples, do not allow for drawing definitive conclusions or make clinical policy recommendations.

Additional data are expected from ongoing clinical trials. The DICIPLE (Double Immune Checkpoint Inhibitors in PD-L1–Positive Stage IV NSCLC [ClinicalTrials.gov identifier: NCT03469960]) phase 3 clinical trial of patients with advanced NSCLC will further assess issues regarding nivolumab-ipilimumab treatment duration. The objective of the study is to demonstrate that a treatment of 6 months, followed by observation (stop and go), is not less effective than a treatment given until progression or toxicity. This strategy is designed with the aim of improving the quality of life of the patients and decreasing the costs and the potential accumulated toxicities. In addition, the SAVE study is evaluating the stop-and-go strategy in advanced NSCLC enrolling patients who responded to anti-PD-1 agents for over a year, comparing OS in those who stopped therapy in comparison to those who continue (Japan Registry of ClinicalTrials identifier: jRCT1031190032). Moreover, ancillary study is planned to examine the prognostic and predictive role of circulating tumor DNA in identifying patients more likely to benefit from therapy discontinuation [33].

## 4. Impact of Rechallenge Treatments

As mentioned before, the therapeutic responses to immunotherapy can be durable after treatment discontinuation, but the progression of the disease can also occur. The uncertainty of the clinical evolution and the need to rescue the life of the patients is a “hot topic” for current and future research, especially regarding the investigation of the possibility to “rechallenge” the tumor by restarting the treatment. Rechallenge at disease progression (after the pre-planned number of cycles) has been recently reported to determine a clinical benefit in up to 70% of the cases; nevertheless, the benefit seems to be limited mainly to patients who obtained a response to initial ICIs and underwent the treatment for at least one year [20].

Data regarding retreatment with PD-1 inhibitors as a subsequent therapy and rechallenge with ICIs are limited and mainly provided from clinical trials on nivolumab and pembrolizumab [16,20,34]. Despite some encouraging findings, currently, no definitive conclusions can be drawn, a cause of the differences in treatment definitions and protocols used in these studies. Recently, a real-life survey using data extracted from the French national healthcare database has been performed, with the aim of describing long-treated patients and their clinical course over time [35]. The results showed that, after retreatment with a PD-1 inhibitor following a first course of nivolumab, oncological outcomes were significantly better in patients with a longer duration of the initial treatment. This finding suggests that retreatment with ICIs may be beneficial in this specific setting.

A possible explanation of the role of immunotherapy in controlling the disease after discontinuation lies in the fact that ICIs induce polyclonal and memory-adaptive antitumor immunity to control the clonal heterogeneity of the disease and reset the tumor-host immune interaction [36]. On the other hand, relapse after ICIs discontinuation may be due to the loss of PD-1/PD-L1 inhibition caused by a rapid antibody clearance or due to resistance due to removal of the PD-1/PD-L1 blockade [37].

## 5. Duration of Immunotherapy and Toxicity

Another critical aspect that impacts the completeness and success of immunotherapy is the arousal of toxicities. Immunotherapy in advanced NSCLC usually has less high-grade toxicities when compared with traditional chemotherapy. Nevertheless, patients who received ICIs may have unpredicted systemic toxicities, with some of them being fatal [38]. Toxicities can affect almost all tissues and organs, more frequently the skin, colon, endocrine glands, liver, and lungs; severe or fatal toxicities (like neurological disorders and myocarditis) are less common [39]. Even though immune-related adverse events (irAEs) are generally mild to moderate (grades 1 and 2), clinically considerable grade 3 or 4 irAEs have been described in up to 20% of patients treated with a single agent targeting PD-1 [40]. Grade 3 or 4 irAEs are more common in patients treated with combination immunotherapy regimens of anti-PD-1 and anti-CTLA-4 medications, involving approximately 60% of patients [41]. These side effects may lead to interruptions of ICIs and steroid therapy prescriptions, which could be linked with poorer survival outcomes, especially when used for a long-time [42]. In some instances, the implementation of biological immunomodulatory agents is necessary to manage patients with life-threatening toxicities not responding to corticosteroids.

The predictive value of irAEs in lung cancer has been investigated in several studies [43,44,45]. Haratani et al., in a cohort of 134 patients with advanced or recurrent NSCLC who were treated with second-line or later nivolumab, observed 69 cases of irAE, which were positively associated with both OS and PFS [43]. In another study published by Akamatsu et al., the presence of irAEs induced by ICIs was correlated with clinical responses, considering that more than 60% of responders to ICIs had irAESs [20]. This correlation was also detected in the Keynote/010 study; among 79 patients in the pembrolizumab arm who completed 35 cycles/2 years of treatment in this study, AEs occurred in 66 patients (83.5%) [46]. Further data regarding patients treated with different ICIs are necessary to confirm the association between the arousal of irAEs and improved clinical responses. Currently, the available scientific evidence suggests that manageable irAEs should not be taken as a factor for treatment cessation, but as a sign of a potential oncological response.

Investigators from Keynote/010 trial found that toxicities accumulate over time, reporting that new grade ≥3 toxicities occurred in approximately 10% of new patients every six months, and this is a strong element to consider limiting the duration of therapy [46]. Although the most common side effects generally occur in the first 5–15 weeks of treatment, existing data demonstrate the onset of late toxicities, both during and after discontinuation of immunotherapy [47,48,49]. In some cases, they can occur even months or years after treatment cessation [50,51,52]. Couey et al., underlined that almost half of the late irAE cases occurred in patients who had already reported previous on-treatment irAEs, often representing the reason for therapy cessation [49]. However, late irAE are not common; Couey et al. found only 21 cases of delayed irAE correlated to ICIs in a ten-year literature data analysis [52]. Nevertheless, they can be both severe and fatal. In a retrospective study by Shah et al., performed on 325 patients treated with ICIs, 12% of the patients who continued therapy for more than 1 year had delayed irAE [53].

## 6. Financial Aspects

The increasing cost of new cancer therapies represents a critical issue in terms of accessibility to cancer healthcare, and decreasing these costs represents currently one of the most challenging public healthcare issues worldwide. Immunotherapy treatment for NSCLC in the United States costs over $100,000 per single patient per year, and when a combination of drugs is employed, over $200,000 per year [54]. To control the economic burden of immunotherapy, several actions have been suggested. Firstly, adequate molecular testing for patient selection should be used; extending the use of multigenic molecular panels to all candidate patients could improve adherence to recommendations and therapeutic strategy planning. In addition, when possible, dose reductions should be taken into consideration. The FDA and the EMA approved ICIs with a flat dose according to exposure-response and pharmacokinetic data. If a life-long treatment is considered, the flat dose compared to the personalized one (related to the patient’s body weight) determines an extra drug dose of around 25–40% and a 25% increase in drug costs [55]. Similar considerations can be made regarding the number of doses and the global duration of the treatments, especially in relation to the occurrence of side effects and the quality of life (QOL) of the patients. ICI de-escalation, other than impacting the toxicity and QOL of the patients, also positively impacts the cost of care. Furthermore, patients who respond well to immunotherapy are treated for an undetermined period, lasting, in some cases, several years, often dictating both physical and financial toxicity. More pharmacoeconomic studies are warranted to better define strategies for cost reduction in this setting.

A 2020 National Cancer Institute study based on retrospective data estimated the annual cost of cancer care in the USA to be over US $200 billion in 2020, and it is expected that it will approach $250 billion by 2030 [56]. Therefore, the search for cost-effective therapeutic strategies is necessary to reduce health cancer costs that are growing dramatically. Immunotherapy has been proven to be cost-effective compared to chemotherapy in advanced lung cancer. In this regard, interesting data emerged from the Empower-Lung 1 study, in which the costs of cemiplimab in first-line treatment of NSCLC patients with at least 50% PD-L1 expression were compared to those in patients treated with chemotherapy, showing that cemiplimab was a cost-effective option [57]. Furthermore, when the pharmacological costs of the drugs were compared based on their efficacy expressed in terms of OS, pembrolizumab and atezolizumab were shown to be cost-effective in first and second-line treatment of metastatic NSCLC patients in comparison to traditional treatments [58].

## 7. Future Perspectives and Conclusions

We currently live in the “immune-revolution” era, where immunotherapy is crucial for the treatment of the most incident cancers worldwide, including NSCLC. Numerous immunotherapy medications are currently available, alone or in combination with both “traditional” chemotherapy medications and other treatments, like radiotherapy or targeted therapies. Further drugs like antiangiogenic agents, immune modulators, PARP inhibitors, and cancer vaccines are nowadays under constant investigation. This plethora of novel therapies is the basis for future studies on a combination of synchronous or heterochronous therapies able to maintain long-term results. In other words, the optimal duration of immunotherapy needs to be established not only in the “first round” but also in the “second or subsequent rounds”, often performed in combination with other medications. Consequently, it will be essential to design trials with a range of immunotherapy combination strategies with other treatments to establish the oncological outcomes and optimal treatment duration in the future.

What we know today is that life-long immunotherapy treatments seem to be unnecessary, expensive, and even harmful in patients with NSCLC; on the other hand, discontinuation of immunotherapy treatment after two years in patients with CR or PR is not yet universally accepted. Current data show that a fixed therapy duration may provide durable clinical benefits, but the costs and toxicities need to be further investigated to establish optimal time scales. In addition, biomarkers that can predict oncological outcomes before or during immunotherapy and help select patients are currently lacking and need to be investigated; particularly interesting in this regard may be studies on PET/CT imaging analyzed with artificial intelligence approaches or consecutive liquid biopsies with wide sequencing of the extracted nucleic acids in search of potential outcome predictors [59].

## Figures and Tables

**Figure 1 cancers-15-00689-f001:**
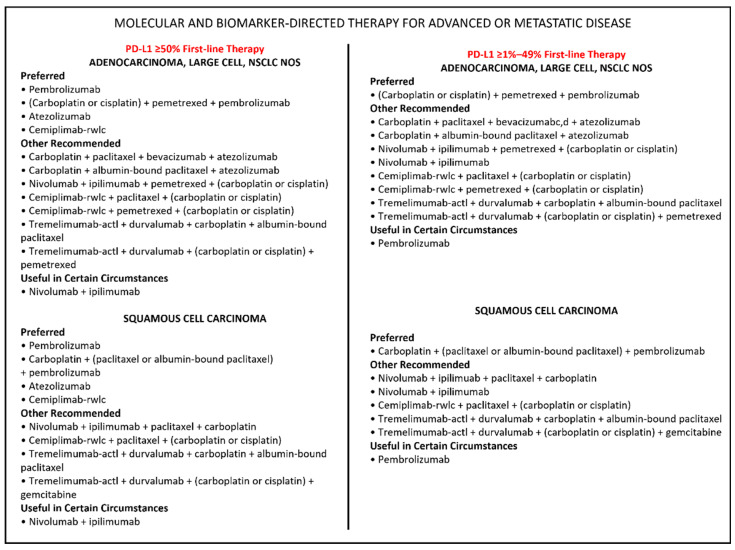
Immunotherapy treatments currently recommended for advanced NSCLC.

**Table 1 cancers-15-00689-t001:** Clinical trials reporting data on immunotherapy durations in patients with NSCLC.

Trial	Phase	Setting	Agent	Treatment Duration
CheckMate 003	Phase 1	Pretreated	Nivolumab	Until progression, unacceptable toxicity, or up to 96 weeks
CheckMate 057	Phase 3	Pretreated	Nivolumab	Until progression, or unacceptable toxicity
CheckMate 017	Phase 3	Pretreated	Nivolumab	Until progression, or unacceptable toxicity
CheckMate 078	Phase 3	Pretreated	Nivolumab	Until progression, or unacceptable toxicity
CheckMate 227	Phase 3	First line	Nivolumab + ipilimumab	Until progression, unacceptable toxicity, or up to 24 months
CheckMate 9LA	Phase 3	First line	Nivolumab + ipilimumab	Until progression, unacceptable toxicity, or up to 24 months
KEYNOTE-001	Phase 1	Any line	Pembrolizumab	Until progression, or unacceptable toxicity
KEYNOTE-010	Phase 2/3	Pretreated	Pembrolizumab	Until progression, unacceptable toxicity, or up to 24 months
KEYNOTE-024	Phase 3	First line	Pembrolizumab	Until progression, unacceptable toxicity, or up to 24 months
KEYNOTE-042	Phase 3	First line	Pembrolizumab	Until progression, unacceptable toxicity, or up to 24 months
KEYNOTE-189	Phase 3	First line	Pembrolizumab	Until progression, unacceptable toxicity, or up to 24 months
POPLAR	Phase 2	Pretreated	Atezolizumab	Until progression, or unacceptable toxicity
OAK	Phase 3	Pretreated	Atezolizumab	Until progression, or unacceptable toxicity
IMpower 130	Phase 3	First line	Atezolizumab	Until loss of clinical benefit or unacceptable toxicity
IMpower 150	Phase 3	First line	Atezolizumab	Until progression, or unacceptable toxicity
MISTIC	Phase 3	First line	Durvalumab (+ tremelimumab)	Until progression, or unacceptable toxicity

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
