# Peer review of "Duration of Immunotherapy in Non-Small Cell Lung Cancer Survivors: A Lifelong Commitment?"

_cancers, 2023, doi:10.3390/cancers15030689_

Round 1

Reviewer 1 Report

It would be interesting to know if, during immune therapy and in case of  disease worsening, you found genetic changes in the neoplastic cells. 

I think that the immune defenses of the patient do not change: you could confirm this. 

Author Response

We would like to thank the reviewer for his/her valuable suggestions. Corrections are highlighted in yellow in the text and listed here point by point.

Issue 1

It would be interesting to know if, during immune therapy and in case of disease worsening, you found genetic changes in the neoplastic cells. I think that the immune defenses of the patient do not change: you could confirm this. 

Reply

We agree that the immune defences do not intrinsically change; immunotherapy agents just have the ability to remove some inhibitions which involve immunological responses to cancer cells. Nevertheless, a series of complex factors, like tumoral heterogeneity, clonality and others impact responses to immunotherapy. These concepts were added in the “Introduction”.  

Reviewer 2 Report

The aim of this Review was to discuss all the aspects related to the optimal duration of immunotherapy in long survivors with non-small cell lung cancer (NSCLC). The Authors presented results of the analysis of the literature indicating that extensive research is needed to understand the real potential of of immunotherapy of NSCLC.

This is a novel and interesting Review that can be published.

There are several small recommendations:

1 Abstract: To attract more readers, please, underline novel insights provided by the Review and….. Please, outline the direction where to move.

2 Please, re- read the text. There are a lot of small typos.

3 It would good to prepare 2 figures of tables explaining Chapters two and three

Author Response

We would like to thank the reviewer for his/her valuable suggestions. Corrections are highlighted in yellow in the text and listed here point by point.

Issue 1

Abstract: To attract more readers, please, underline novel insights provided by the Review and….. Please, outline the direction where to move.

Reply

The abstract was modified in accordance with the reviewer’s suggestions.

Issue 2

2 Please, re- read the text. There are a lot of small typos.

Reply

Extensive English editing has been performed.

Issue 3

3 It would good to prepare 2 figures of tables explaining Chapters two and three

Reply

A table and a figure have been added in the text.

Reviewer 3 Report

Abstract: Line 28 check for grammatical edits

Introduction Line 47 Global Cancer Observatory

The Introduction section has very sparse referencing with prominent reference

Impact segment is well explained. Authors can consider use of relevant references

Future perspectives and conclusions: Line 349: The meaning is not clear.

The transition from one section tot he next is missing.

The authors can consider explaining the concept/summary via a schematic diagram.

The conclusion is vague and needs to be more scientifically sound.

Author Response

We would like to thank the reviewer for his/her valuable suggestions. Corrections are highlighted in yellow in the text and listed here point by point.

Issue 1
Abstract: Line 28 check for grammatical edits

Reply

Extensive English editing has been performed. Grammatical errors and typos have been corrected.

Issue 2

Introduction Line 47 Global Cancer Observatory

Reply

Extensive English editing has been performed. Grammatical errors and typos have been corrected.

Issue 3

The Introduction section has very sparse referencing with prominent reference Impact segment is well explained. Authors can consider use of relevant references

Reply

Additional references have been added to the “Introduction”.

Issue 4

Future perspectives and conclusions: Line 349: The meaning is not clear.

Reply

The sentence has been removed. All the Future perspectives and conclusion section was revised.

Issue 5

The transition from one section tot he next is missing.

Reply

Transition from one section to the next has been improved.

Issue 6

The authors can consider explaining the concept/summary via a schematic diagram.

Reply

A table and a figure have been added to the text.

Issue 7

The conclusion is vague and needs to be more scientifically sound.

Reply

The “Conclusions” section has been revised.